# Assessing the Validity and Reliability of a French Version of the Child Feeding Questionnaire among Black Mothers Living in Ottawa (Canada)

**DOI:** 10.3390/bs13060487

**Published:** 2023-06-09

**Authors:** Alexandra M. Bodnaruc, Cris-Carelle Kengneson, Rosanne Blanchet, Isabelle Giroux

**Affiliations:** 1School of Nutrition Sciences, Faculty of Health Sciences, University of Ottawa, Ottawa, ON K1N 6N5, Canada; abodn049@uottawa.ca (A.M.B.); ckeng055@uottawa.ca (C.-C.K.); 2Department of Social and Preventive Medicine, School of Public Health, Université de Montréal, Montreal, QC H3T 1J4, Canada; rosanne.blanchet@umontreal.ca

**Keywords:** Child Feeding Questionnaire, validity, reliability, confirmatory factor analysis, French CFQ, Black mothers

## Abstract

The Child Feeding Questionnaire (CFQ) is a well-established tool used to assess parental beliefs, attitudes, and child feeding practices, with a focus on childhood obesity proneness. To date, there is no French version of the CFQ, nor any Canadian studies assessing its construct validity. The aim of this study was to assess the construct validity and the reliability of a French version of the CFQ among Black mothers (*n* = 136) of school-aged children living in Ottawa (Canada). The final best fitting model included 7 factors, 20 items, and 1 error covariance. This model was retained as the final model as it (1) excluded two items with very low factor loadings; (2) had the lowest χ^2^, AIC, BIC, RMSEA, and SRMR values; and (3) had CFI and TLI values ≥ 0.95. Internal consistency ranged from poor to good; the restriction subscale had the lowest internal consistency, followed by the perceived responsibility, pressure to eat, perceive child weight, concern about child weight, and monitoring scales, respectively. Our results showed that a seven-factor model with minor modifications was best fitted to the current data. Future studies are needed to test the validity and reliability of the CFQ in other population groups and among fathers.

## 1. Introduction

Childhood obesity is a growing public health problem involving both immediate and long-term consequences on physical (e.g., higher risk of developing type 2 diabetes, cardiovascular disease) and mental (e.g., lower self-esteem, higher risk of developing depression) health [1,2]. It is well-established within the scientific literature that, due to differences in genetic backgrounds, health-related behaviors, and socioeconomic status, the prevalence of excess weight varies across racial and ethnic groups [3,4]. In Canada, Black individuals are among the highest-risk population groups, with overweight and obesity rates nearing 50% and 29% among adults and children (2–17 years old), respectively [3,5,6].

Socioecological factors such as family have been shown to influence childhood obesity risk [7,8]. According to Anderson et al., “family is the first and most fundamental socio-environmental context in which eating patterns are established” [9]. Through shaping the family’s food environment, parents influence their child’s eating behaviors and perhaps their lifelong weight trajectories [10,11]. Studies assessing parental child feeding practices have reported that controlling child feeding practices such as restriction and pressure to eat could interfere with the child’s ability to recognize internal hunger and satiety cues, which could eventually impact the child’s weight status [12,13,14,15]. In this regard, several cross-sectional and longitudinal studies have reported that parents who use restrictive feeding practices (i.e., restricting their children’s access to foods, particularly “unhealthy” foods) were more likely to have children with a higher body mass index [12,13,14,15,16,17,18]. Conversely, parental use of pressure to eat (i.e., pressuring their children to eat more food, typically at mealtimes) was predominately found in children with a lower BMI [12,13,14,15,16,17,18]. Parental use of monitoring practices (i.e., overseeing their child’s eating) was not associated with child’s BMI in most studies [12,13,14,15,16,17,18]. Some longitudinal studies further suggest that parents’ perceptions of their child’s weight status influence the child’s weight over time [19,20,21]. These studies show that children whose parents perceived them as having overweight or obesity were more likely to gain and maintain excess weight over time [19,20,21]. Understanding parents’ child feeding practices and how they relate to children’s present and future dietary intakes, eating behaviors, and weight status is important as it might help identify ways to improve overweight and obesity prevention and intervention strategies [12,13,14,15].

For over twenty years, research on child feeding practices has been conducted using various validated tools including the Child Feeding Questionnaire (CFQ) [22,23,24]. The CFQ is a well-established and validated tool used to assess parental beliefs, attitudes, and practices regarding child feeding, with a focus on obesity proneness in children [25]. The CFQ comprises four subscales assessing parental perceptions and concerns about weight status (i.e., perceived parent’s weight, perceived child’s weight, parental concern about child’s weight, and perceived parental responsibility) and three subscales assessing parental control attitudes and practices in child feeding (restriction, pressure to eat, and monitoring). This questionnaire was developed by Birch et al. (2001) and initially validated on a mainly White sample [25]. Afterwards, several researchers have explored the questionnaire’s psychometric properties and subscales’ structure in various populations such as Hispanic American [9,26], African American [9,26,27], Australian [28], Japanese [29], Turkish [30], Swedish [31], Chinese [32], Spanish [33], German [34], and Arab [35] populations (see Table 1). While acceptable modified structures were presented in previous validation studies, the conceptualization and measurement of some subscales, such as restriction, perceived child’s weight, and perceived parent’s weight, have been questioned in some studies [9,29,30,31,36]. The restriction subscale has been identified in multiple studies to be unstable [9,27,28,29,30,31,33,36]. In fact, some specific items of the restriction subscale appear to be problematic, namely those related to food used as a reward. Several approaches have been used to improve the stability of the restriction subscale, such as omitting some items [9,29,31,36], while others have created a new subscale named “food as a reward’’ with the two items of the restriction subscale [26,28,32,34].

Assessing the cross-cultural and racial equivalence of the CFQ has been pointed out as a pending need in several reviews focusing on measures of parental feeding practices [22,23,24,37]. To date, there is no French version of the CFQ nor any Canadian study assessing its construct validity. Furthermore, there are very few Canadian studies assessing parental beliefs, attitudes, and practices regarding child feeding using the CFQ [38,39,40]. While Black individuals living in Canada have among the highest rates of obesity in the country, these studies have been conducted in predominantly White and English-speaking samples [38,39,40]. Therefore, this study aimed to assess the reliability and construct validity of a French version of the CFQ among Black mothers of school-aged children living in Ottawa (Canada).

## 2. Materials and Methods

### 2.1. Participants 

Mothers in this analytic sample were part of a cross-sectional study aiming to assess the social determinants of nutritional heath of 6-to-12-year-old children of African and Caribbean descent. The study, including recruitment strategies and findings, has been described in several previously published manuscripts [18,41,42,43,44,45]. The study was conducted between January 2014 and April 2015 in Ottawa (Canada). To be eligible for the study, mothers had to have at least one child aged between 6 and 12 years, to be born in sub-Saharan Africa, the Caribbean, or Canada, and be able to understand and speak English or French. Due to genetics being a risk factor for overweight and obesity, mothers of adopted children were excluded. Children who had food allergies occasioning important dietary restrictions, and/or suffered from metabolic or congenital disorders that may influence growth, and/or limited the intake of certain foods, and/or suffered from any medical condition limiting or preventing their participation in normal physical activities were also excluded. When mothers had two or more children eligible to participate in the study, one of them was randomly chosen. Mothers completed the CFQ in either English or French, according to their preference. A total of 252 mothers participated in this study. Two hundred and twelve (*n* = 212) mothers were Black, among which one hundred and forty-one chose to complete the CFQ in French. A total of 5 mothers were excluded because they self-identified with more than one ethnicity, leaving 136 mothers for the current analyses.

The study was approved by the Office of Research Ethics and Integrity of the University of Ottawa (H-02-19-1405). Informed consent and assent were obtained from mothers and children, respectively.

### 2.2. Measures 

#### 2.2.1. Sociodemographic Characteristics

Mothers’ (i.e., age, migration status, length of time since immigration, education level, and marital status) and children’s (i.e., age and sex) socio-demographic characteristics were collected through a questionnaire administered to the mother by a trained research team member. Length of time since migration and age were calculated using the data collection date, along with the reported date of arrival in Canada and mothers’ and children’s dates of births, respectively.

#### 2.2.2. Racial and Ethnic Identity 

Ethnicity refers to the shared understanding of the historical and territorial origins of a population group as well as to a distinctive set of cultural similarities between people [46,47]. The latter concept is flexible and requires self-identification to one or more pre-established and/or self-perceived ethnic groups. In this study, ethnic identity was determined according to participants’ responses to the question: “With which racial or cultural group do you identify? You may belong to one or more racial or cultural groups on the following list: White, South Asian, Chinese, Black, Filipino, Latin American, Arab, Southeast Asian, West Asian, Korean, Japanese, and Other”.

#### 2.2.3. Weight Status 

Mothers’ and children’s height (Charder HM200P Portstad portable stadiometer) and weight (LifeSource ProFit UC-321, A&D Medical digital scale) were measured by a registered dietitian or a trained dietetic student using the World Health Organisation (WHO) guidelines [48]. Mothers’ body mass index (BMI, weight (kg)/height (m)^2^) was calculated and classified according to the WHO’s BMI cut-offs [48]. The International Obesity Task Force (IOTF) references were used to calculate BMI-for-age-and-sex z-scores and define children’s weight status [49]. 

#### 2.2.4. Maternal Feeding Practices, Perceptions, and Concerns about Child’s Weight

The CFQ was used to assess maternal feeding practices, perceptions, and concerns about child’s weight [25]. All of the CFQ’s subscales, subscales’ items, and answer scales are shown in Table 2. Three items of the PPW subscale and three items of the PCW subscale were dropped prior to the study in order to shorten the survey. Items assessing mothers’ perception of their weight during childhood (PPW1), adolescence (PPW2), and her twenties (PPW3) were dropped to shorten the questionnaire and due to the fact that an important proportion (94.1%) of Black mothers in this study were immigrants and thus spent part of their life in a Sub-Saharan African or Caribbean country, where rates of overweight and obesity were low. As there are important variations in body shape standards across cultures and because immigrant mothers came to Canada at different life stages, only mothers’ perception of their current body weight (PPW4) was assessed. As for the PCW subscale, items assessing parents’ perception of their child’s weight after kindergarten (PCW4–6) were dropped. As participating children were aged 6-to-12 years old, only items PCW1–PCW3 could be answered by all mothers, while items PCW4-PCW6 could only be answered by mothers with older children.

### 2.3. Translation of the Child Feeding Questionnaire 

The back-translation method with bilingual testing was used for translating into French the original English version of the CFQ [50]. Two bilingual (French-English) individuals whose first language was French individually completed the English-to-French translation of the CFQ. A panel composed of two bilingual co-investigators of the study compared, discussed, and modified translations to get the first complete French version of the questionnaire. Using the same approach, the French questionnaire was translated back into English by two bilingual individuals whose first language was English. The same panel compared the different English translations as well as the French and English translations, which led to a second version of the questionnaire in both languages. Both the English and the French versions of the questionnaire were tested among bilingual individuals of the target population before some final modifications were made. After the translation process, the response options “seldom”, “markedly underweight”, “markedly overweight”, and “unconcerned” of the original English version of the CFQ were replaced by “rarely”, “very underweight”, “very overweight”, and “not concerned”, respectively, to lower the language level as many participants had low English literacy. The English and the French versions of the questionnaire are presented in Table 2. 

### 2.4. Statistics 

#### 2.4.1. Software and Packages 

Data processing and statistical analyses were computed in R version 4.0.2 for (Mac) OS X [51]. Multivariate imputation via chained equations, item measures and reliability, and confirmatory factor analyses were computed using MICE [52], psych [53], and lavaan [54] packages.

#### 2.4.2. Item Measures 

All items’ and subscales’ mean, standard deviation, skewness, and kurtosis measures were computed as indicators of data distribution. Floor and ceiling effects were assessed and reported when an item was rated with the lowest or highest possible score, respectively, by ≥50% of participants. Spearman’s ρ was computed to assess inter-item as indicators of item redundancy. Spearman’s ρ values ranging between 0.15 and 0.50 for inter-item correlations were considered acceptable [55,56].

#### 2.4.3. Validity 

Confirmatory factor analysis (CFA) was used to evaluate the factor structure of the CFQ using maximum likelihood with robust standard error and a Santa–Bentler scaled test statistic (MLM) estimation [57]. MLM estimation was used as items evaluated on Likert-type scales can be seen as ordinal variables and may violate the assumption of multivariate normality [57]. The structures of all tested models are summarized in Table 3. Composite items present in some models were computed through averaging items scores (i.e., RST1a, RST1b, and RST1c into RST1; RST3a and RST3b into RST3; RST4a and RST4b into RST4).

The adequacy of models’ fit was assessed using the χ^2^ test, Akaike Information Criterion (AIC), Bayesian Information Criterion (BIC), Comparative Fit Index (CFI), Tucker–Lewis Fit Index (TLI), Root Mean Square Error of Approximation (RMSEA), and Standardized Root Mean Square Residual (SRMR). The χ^2^ test compares the correlation matrix of the tested model to that estimated under the hypothesized (or “best fitting”) model or floor effects as well as high kurtosis and skewness values, which may have contributed to the low reliability of those subscales [58]. Small and non-significant χ^2^ values are indicative of small discrepancies between the observed data structure and the data structure of the hypothesized model [58]. AIC and BIC estimate the risk of under-fitting (AIC) and over-fitting (BIC) based on several parameters, including model complexity. Lower AIC and BIC values are associated with a lower risk of under-fitting and over-fitting, respectively [58,59,60]. CFI and TLI compare the tested model to a null (or “worst fitting”) model, with values ranging from 0.90 to 0.94 indicating an acceptable fit and values ≥ 0.95 indicating a good fit [58,61]. RMSEA compares the tested model to a reasonably fitted one, with values ranging from 0.06 to 0.08 indicating an acceptable fit and values < 0.06 indicating a good fit [58,61]. SRMR compares the tested model to a perfectly fitted one, with values nearing 0.08 indicating a good fit [58,61].

#### 2.4.4. Reliability 

Cronbach α and Macdonald’s ω were computed as estimates of subscales’ internal consistency [62,63,64]. Cronbach α and Macdonald’s ω if item deleted were also computed to identify any item that disproportionately impacted reliability estimates.

## 3. Results

### 3.1. Sample Characteristics 

Mothers and children were aged 39.1 (SD: 5.9, range: 28–54) and 8.8 (SD: 2.0, range: 6–12) years, respectively. A majority of mothers were foreign-born (94.1%), had a partner at the time of the study (66.2%), and obtained at least a college certificate or diploma (88.9%). Migrant mothers had been living in Canada for an average of 8.8 (SD: 8.1, range: 0.03–38.6) years. Overweight and obesity prevalence was high in both mothers (82.3%) and children (35.1%). Mothers’ and children’s characteristics are shown in Table 4.

### 3.2. Item Measures 

Items’ and subscales’ mean, standard deviation, distribution, and reliability estimates are shown in Table 5. There were less than 1% missing data. Skewness values were acceptable (skew = −1.82–1.17) for the large majority of items and subscales, with only the PR3 item being negatively skewed (skew = −2.27). Several items (PR1, PR3, PCW1–3, RST1, PE2) and one subscale (PCW) had a leptokurtic distribution, as indicated by kurtosis values > 2.0. All high kurtosis values were positive, which is consistent with the fact that ceiling effects were observed for seven items; six items (PR1, PR3, RST1, RST2, RST4, PE4) were rated with the highest possible score by 50–74% of participants, and one item (PE2) was rated with the highest possible score by ≥75% participants. Floor effects were observed for CCW1 and CCW2, with these items being rated with the lowest possible score by 50–55% of participants.

### 3.3. Validity 

Fit indices for all tested models are shown in Table 3. Out of the nine tested models, three (model 3, model 6, and model 9) had an acceptable fit as indicated by non-significant χ^2^ statistic, CFI ≥ 0.95, TLI ≥ 0.95, RMSEA ≤ 0.05, and SRMR ≤ 0.08. Of these three acceptable models, model 6 (see Figure 1) was retained as the final model as it (1) excluded two items with very low factor loadings; (2) had the lowest χ^2^, AIC, BIC, RMSEA, and SRMR values; and (3) had CFI and TLI values ≥ 0.95. The final model, presented in Figure 1, included 7 factors, 20 items, and 1 error covariance between PCW1 and PCW2. All factor loadings were significantly different from 0 and were ≥0.30 (see Table 6), inter-item correlations ranged from 0.10 to 0.50, and inter-factor correlations ranged from 0.001 to 0.613. There were several significant inter-factor correlations. MN was positively correlated with PR (r = 0.37, *p* < 0.05), CCW (r = 0.28, *p* < 0.01), RST (r = 0.61, *p* < 0.01), and PE (r = 0.33, *p* < 0.01), while PE was positively correlated with PR (r = 0.28, *p* < 0.05) and RST (r = 0.36, *p* < 0.01) and negatively correlated with PPW (r = −0.26, *p* < 0.05).

### 3.4. Reliability 

With the final model’s (model 6) structure, internal consistency ranged from poor (Cronbach α = 0.53, Macdonald’s ω = 0.60) to good (Cronbach α = 0.80, Macdonald’s ω = 0.82) (see Table 4). The RST subscale had the lowest internal consistency (Cronbach α = 0.53, Macdonald’s ω = 0.53), followed by the PR (Cronbach α = 0.57, Macdonald’s ω = 0.60), PE (Cronbach α = 0.64, Macdonald’s ω = 0.70), PCW (Cronbach α = 0.68, Macdonald’s ω = 0.78), CCW (Cronbach α = 0.79, Macdonald’s ω = 0.79), and MN (Cronbach α = 0.80, Macdonald’s ω = 0.82) scales, respectively. Removing any of the three items of model 6’s RST subscale did not increase internal consistency. Including the composite item RST3 in the RST subscale was associated with a slightly poorer internal consistency (Cronbach α = 0.51, Macdonald’s ω = 0.57). On the contrary, models without composite items on the RST scale had higher internal consistency (Cronbach α = 0.66, Macdonald’s ω = 0.79). Similar to the RST scale, removing any item from the PR subscale did not improve internal consistency. For the two subscales with an acceptable internal consistency, removing an item increased Cronbach α values. Removing item PE1 from the PE subscale increased the Cronbach α value from 0.64 to 0.70, and removing item PCW3 from the PCW subscale increased the Cronbach α value from 0.68 to 0.73. The RW subscale that was only included in models 7, 8, and 9 had poor internal consistency (Cronbach α = 0.32, Macdonald’s ω = 0.60).

## 4. Discussion 

This study aimed to develop a French version of the CFQ and to assess its construct validity and reliability in a sample of Black mothers of 6-to-12-year-old children living in Canada. In addition to the English-to-French translation, six items (PPW1–3, PCW3–6) were removed from the original version of the CFQ prior to its administration. Based on previous validation studies, several CFQ structures were tested, among which a seven-factor model, with minor modifications, was found to have the best fit to the current data. Most modifications made to the questionnaire’s structure involved the RST subscale, whose construct limitations were highlighted in several previous studies. Except for RST and PR, all subscales had at least an acceptable level of reliability.

Similar to Schmidt et al. (2017), item analyses revealed a low number of missing data indicating that, when translated into French, the questionnaire was comprehensible overall and well-accepted by the mothers. A leptokurtic distribution and ceiling effects were noted for several items, namely, items of the PR, RST, and PE scales. While floor and ceiling effects are more likely to be observed in population-based samples [34,65], these findings raise concerns over the sensitivity of some factors of the CFQ. Since only Schmidt et al. (2017) have reported item measures, additional studies doing so are needed in order to adequately document the sensitivity of the CFQ in various population-based and clinical (e.g., treatment-seeking) samples.

In the present study, modifications made to the questionnaire during the validation process reflected previously encountered challenges. The factor causing the most problems in all validation studies is RST. While most of the items loaded as predicted and had acceptable loading factors, the composite item “food as reward” loaded poorly on the RST subscale. In the large majority of previous validation studies, these two items were either dropped [9,31,36] or used to create a new subscale entitled “food as reward” [26,32,34]. Here, we tested both approaches and retained the model in which the two RST items were dropped (Model 6, Table 3) rather than the eight-factor model (Model 9, Table 3). While both models had an acceptable construct, the retained model had better fit indices. Overall, the final model’s fit indices were better than those reported in previous studies (see Table 1 for a summary of previous studies’ fit indices). 

Another problematic factor was PCW, which had high inter-item correlations. As previously pointed out, the repetitive nature of the questions combined with a relatively short age span may have led to strong correlations between PCW items [31,36]. To address this issue, we included one error covariance between PCW1 and PCW2 in the final model. Most previous studies, including Birch et al. (2001), also included error covariances between items of the PCW subscale in their final model to address this problem [25,31,33,34,36]. 

Our results showed that PCW, CCW, PE, and MN subscales had good or acceptable reliability, while PR and RST subscales had low reliability. Both the PR and the RST subscale included several items with either ceiling or floor effects as well as high kurtosis and skewness values, which may have contributed to the low reliability of those subscales [63,64]. Overall, reliability was weaker than in previous studies, which may partly be explained via differences in sample sizes. Larger sample sizes and continuous data allow for a more valid estimate of reliability [62,63,64]. 

This is the first study to assess the reliability and construct validity of a French translation of the CFQ. The results of this study should be considered in light of several limitations. This study was conducted in a relatively small sample size of Black mothers of 6-to-12-year-old children living in Ottawa, which limits its generalizability to other racial and ethnic groups as well as to Black mothers living outside Ottawa. Another limitation of the present study is the fact that only one item from the PPW subscale was included in the questionnaire administered in the present study. Previous studies have reported high correlations between PPW items, which affect the fit indicators of the tool. The absence of these factors may have contributed to improving the fit indicators in this study. Future studies are needed to test the validity and reliability of the questionnaire among fathers, among individuals from other racial and ethnic groups, as well as among individuals living in different regions of Canada.

## Figures and Tables

**Figure 1 behavsci-13-00487-f001:**
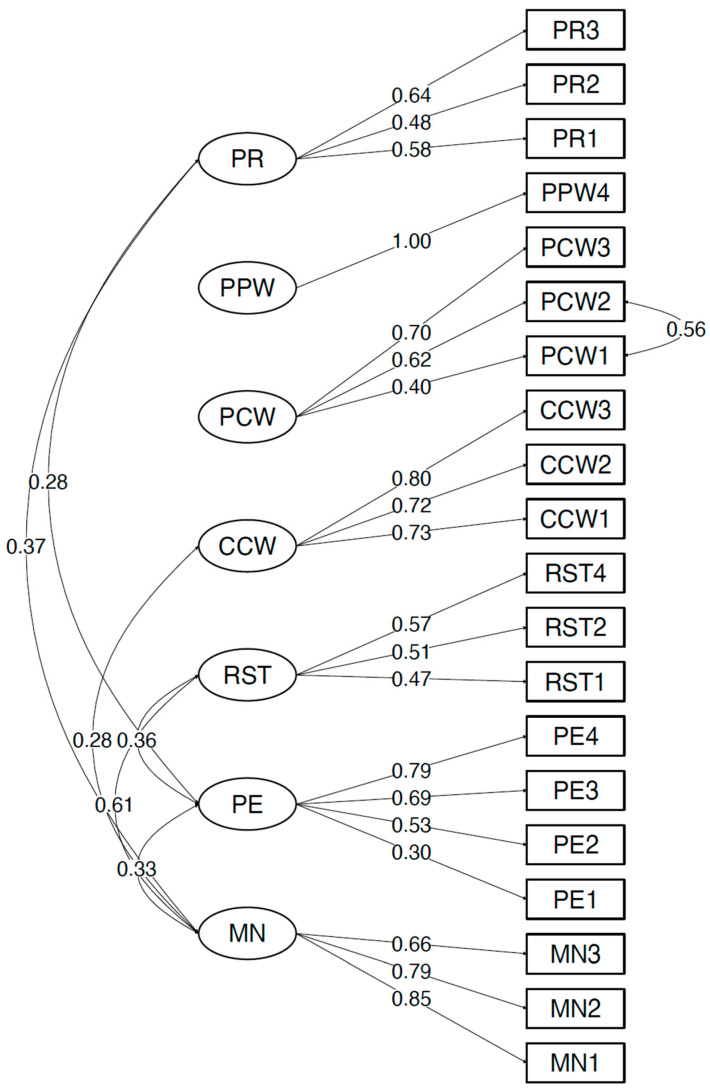
Model 6 factorial structure. CCW: Concern about Child’s Weight; MN: Monitoring; PCW: Perceived Child Weight; PE: Pressure to Eat; PPW: Perceived Parent Weight; PR: Perceived Responsibility; RST: Restriction.

**Table 1 behavsci-13-00487-t001:** Characteristics of best-fitting models in previous studies assessing CFQ’s construct validity using CFA.

Authors (Year)	Population	Language	Characteristics of the Best FITTING Model (s)							
			Factors	Items	Restriction Scale	Other	Fit Indices			
Birch et al. (2001) [25]	Sample 1: Non-Hispanic White (mostly) parents of 5–9-year-old girls living in the US (*n* = 394)	English	7	24	RST1a,b,c (avg) RST2 RST3a,b (avg) RST4a,b (avg)	PPW1 ~~ PPW2 PCW1 ~~ PCW2	χ^2^ = 419 *	AIC = nr BIC = nr	CFI = 0.95 TLI = 0.94	RMSEA = 0.04 SRMR = nr
	Sample 2: Non-Hispanic White (mostly) parents of 8–11-year-old children living in the US (*n* = 148)					RSTa,b ~~ PE PPW1 ~~ PPW3	χ^2^ = 309		CFI = 0.92 TLI = 0.91	RMSEA = 0.05 SRMR = nr
	Sample 3: Hispanic (mostly) parents of 7–11-year-old children living in the US (*n* = 126)			21		Items dropped or omitted: PE1, PE2, RST3a,b	χ^2^ = 232		CFI = 0.91 TLI = 0.89	RMSEA = 0.05 SRMR = nr
Anderson et al. (2005) [9]	Parents of 3-year-old Black and Hispanic children living in the US (*n* = 231)	English	5	16	RST1a RST1b RST4a	Scales dropped or omitted: PPW, PCW Items dropped or omitted: RST1c, RST2, RST3a,b, RST4b	χ^2^ = 129 **	AIC = nr BIC = nr	CFI = 0.96 TLI = 0.95	RMSEA = 0.04 SRMR = 0.06
	Parents of 3-year-old Black children living in the US (*n* = 101)						χ^2^ = 71		CFI = 0.93 TLI = 0.92	RMSEA = 0.05 SRMR = 0.09
	Parents of 3-year-old Hispanic children living in the US (*n* = 130)						χ^2^ = 167		CFI = 1.00 TLI = 1.04	RMSEA = 0.00 SRMR = 0.06
Kaur et al. (2006) [36]	Parents of 10–19-year-old ethnically diverse adolescents living in the US (*n* = 260)	English	7	24	RST1a,b,c (avg) RST2 RST4a,b (avg)	PPW1 ~~ PPW2 PCW1 ~~ PCW2	χ^2^ = 357	AIC = nr BIC = nr	CFI = 0.95 TLI = 0.93	RMSEA = 0.05 SRMR = nr
Corsini et al. (2008) [28]	Mothers of 4- to 5-year-old children living in Australia (*n* = 203)	English	7	24	RST1a,b,c (avg) RST2 RST3a,b (avg) RST4a,b (avg)	Items dropped or omitted: PCW4, PCW5, PCW6	χ^2^ = 399 **	AIC = 537 BIC = 766	CFI = 0.93 TLI = 0.92	RMSEA = 0.06 SRMR = nr
Geng et al. (2009) [29]	Parents of 9- to 12-year-old children living in Japan (*n* = 920)	Japanese	7	24	RST1a,b,c (avg) RST2 RST3a,b (avg) RST4 removed	PPW1 ~~ PPW2 PPW2 ~~ PPW3 PPW3 ~~ PPW4 Items dropped or omitted: PCW1, PCW6	χ^2^ = 1115 **	AIC = nr BIC = nr	CFI = 0.93 TLI = 0.92	RMSEA = 0.06 SRMR = 0.05
Liu et al. (2014) [32]	Chinese-Australian mothers of 1- to 4-year-old children (*n* = 254)	Chinese	8	27	RST1a,b,c RST2 RST4a,b	Items dropped or omitted: PCW3, PCW4, PCW5, PPW3 RW scale: RST3a, RST3b	χ^2^ = nr	AIC = 763 BIC = nr	CFI = 0.90 TLI = 0.88	RMSEA = 0.06 SRMR = nr
Nowicka et al. (2014) [31]	Mothers of 4-year-old children living in Sweden (*n* = 564)	Swedish	7	23	RST1a,b,c (avg) RST2 RST4a,b (avg)	Items dropped or omitted: PCW4, PCW5, RST3a, RST3b PPW1 ~~ PPW2 PCW1 ~~ PCW2 PE3 ~~ PE4	χ^2^ = 345	AIC = nr BIC = nr	CFI = 0.96 TLI = 0.94	RMSEA = 0.04 SRMR = 0.05
Kong et al. (2014) [26]	Low-income Hispanic and African American mothers of 2- to 5-year-old children (*n* = 962)	English	6	20	RST1a,b,c RST2 RST4a,b	Scales dropped or omitted: PCW, PPW Items dropped or omitted: PR1, PR4 RW scale: RST3a, RST3b	χ^2^ = 686 **	AIC = nr BIC = nr	CFI = 0.95 TLI = 0.94	RMSEA = 0.06 SRMR = nr
Canals-Sans et al. (2016) [33]	Parents of 6- to 10-year-old children living in Spain (*n* = 960)	Spanish	7	29	8 items	Items dropped or omitted: PCW5, PCW6 RST1a ~~ RST1b PCW2 ~~ PCW3 PCW1 ~~ RST3a PCW1 ~~ RST3b	χ^2^ = 766 **	AIC = nr BIC = nr	CFI = 0.94 TLI = 0.94	RMSEA = 0.04 SRMR = 0.05
Schmidt et al. (2017) [34]	Mothers of 2- to 13-year-old children living in Germany (*n* = 982)	German	8	26	RST1a,b,c RST2 RST4a,b	Items dropped or omitted: PCW4, PCW5 RW scale: RST3a, RST3b PPW1 ~~ PPW2 PPW2 ~~ PPW3 PPW3 ~~ PPW4 PCW1 ~~ PCW2 PCW2 ~~ PCW3	χ^2^ = 729 *	AIC = 945 BIC = nr	CFI = 0.96 TLI = 0.94	RMSEA = 0.05 SRMR = na
Mosli (2020) [35]	Mothers of pre-school children living in Saudi Arabia (*n* = 209)	Arabic	7	29	8 items	-	χ^2^ = 257 **	AIC = nr BIC = nr	CFI = 0.97 TLI = 0.96	RMSEA = 0.02 SRMR = 0.07
Present paper	Black mothers of 6- to 12-year-old children living in Canada (*n* = 136)	French	7	21	RST1a,b,c RST2 RST4a,b	Items omitted: PCW4, PCW5, PPW1–3 Items dropped: RST3a, RST3b	χ^2^ = 170 (151)	AIC = 7754 BIC = 7734	CFI = 0.96 TLI = 0.96	RMSEA = 0.03 SRMR = 0.06

~~: error covariance. * *p* < 0.05. ** *p* < 0.01. AIC: Akaike Information Criterion; AVG: Average; BIC: Bayesian Information Criterion; CFA: Confirmatory Factor Analysis; CFI: Comparative Fit Index; CFQ: Child Feeding Questionnaire; NR: Not Reported; PCW: Perceived Child Weight; PE: Pressure to Eat; PPW: Perceived Parent Weight; PR: Perceived Responsibility; RMSEA: Root Mean Square Error of Approximation; RST: Restriction; RW: Reward; SRMR: Standardized Root Mean Square Residual; TLI: Tucker–Lewis Fit Index.

**Table 2 behavsci-13-00487-t002:** English version and French translation of the Child Feeding Questionnaire.

Items—English Version	Items—French Translation	Answer Choices	
1. Perceived responsibility (PR)	1. Responsabilité perçue		
PR1: When your child is at home, how often are you responsible for preparing his/her meals? PR2: How often are you responsible for deciding what your child’s portion sizes are? PR3: How often are you responsible for deciding if your child has eaten the type/right kind of food?	PR1: Lorsque votre enfant est à la maison, à quelle fréquence êtes-vous responsable de le/la nourrir ? PR2: À quelle fréquence êtes-vous responsable de décider la grosseur des portions de votre enfant ? PR3: À quelle fréquence êtes-vous responsable de décider si votre enfant a mangé les bons types d’aliments ?	1 = Never 2 = Rarely 3 = Half of the time 4 = Most of the time 5 = Always	1 = Jamais 2 = Rarement 3 = La moitié du temps 4 = La plupart du temps 5 = Toujours
2. Perceived parent weight (PPW)	2. Poids perçu du parent		
How would you rate your weight? PPW1: During childhood (5 to 10 y) PPW2: During adolescence PPW3: In your 20′s PPW4: Now	Comment classifieriez-vous votre poids ? PPW1: Pendant l’enfance (5 à 10 ans) PPW2: Pendant l’adolescence PPW3: Pendant la vingtaine PPW4: Présentement	1 = Very underweight 2 = Underweight 3 = Normal 4 = Overweight 5 = Very overweight	1 = Très maigre (sous-poids) 2 = Maigre (sous-poids) 3 = Normal 4 = En surpoids 5 = Très en surpoids
3. Perceived child’s weight (PCW)	3. Poids perçu de l’enfant		
How would you rate your child’s weight at these stages? PCW1: First year of life PCW2: Toddler (1 to 2 y) PCW3: Kindergarten (3 to 5 y) PCW4: Kindergarten–2nd grade PCW5: 3rd–5th grade PCW6: 6th–8th grade	Comment classifierez-vous les poids de votre enfant lors de ces périodes ? PCW1: Première année de vie PCW2: Lorsqu’il avait 1–2 ans (bambin) PCW3: Lorsqu’il avait 4–5 ans (pendant la maternelle et le jardin d’enfant) PCW4: Entre la maternelle et la 2e année PCW5: Entre la 3e et la 5e année PCW6: Entre la 6e et la 8e année	1 = Very underweight 2 = Underweight 3 = Normal 4 = Overweight 5 = Very overweight	1 = Très maigre (sous-poids) 2 = Maigre (sous-poids) 3 = Normal 4 = En surpoids 5 = Très en surpoids
4. Concern about child’s weight (CCW)	4. Préoccupation au sujet du poids de l’enfant		
CCW1: How concerned are you about your child eating too much when you are not around? CCW2: How concerned are you that your child will have to diet to maintain a desirable weight? CCW3: How concerned are you about your child being overweight in the future?	CCW1: À quel point êtes-vous préoccupée par le fait que votre enfant mange trop lorsque vous n’êtes pas là ? CCW2: À quel point êtes-vous préoccupée par l’idée que votre enfant devra suivre un régime ou une diète pour maintenir un poids désirable ? CCW3: À quel point êtes-vous préoccupée par l’idée que votre enfant soit en surpoids dans le futur ?	1 = Not concerned 2 = A little concerned 3 = Concerned 4 = Fairly concerned 5 = Very concerned	1 = Pas préoccupée 2 = Un peu préoccupée 3 = Préoccupée 4 = Assez préoccupée 5 = Très préoccupée
5. Restriction (RST)	5. Restriction		
RST1a: I have to make sure that my child does not eat too many sweets (candy, ice-cream, cake, or pastries). RST1b: I have to make sure that my child does not eat too many high-fat foods. RST1c: I have to make sure that my child does not eat too many of his/her favorite foods. RST2: I intentionally keep some foods out of my child’s reach. RST3a: I offer sweets (candy, ice-cream, cake, or pastries) to my child as a reward for good behavior. RST3b: I offer my child his/her favorite foods as a reward for good behavior. RST4a: If I did not guide or regulate my child’s eating, he/she would eat too many junk food (pizza, fries, poutine, hot-dogs). RST4b: If I did not guide or regulate my child’s eating, he/she would eat too many of his favorite foods.	RST1a: Je dois m’assurer que mon enfant ne mange pas trop de sucreries (bonbons, crème glacée, gâteau ou pâtisserie). RST1b: Je dois m’assurer que mon enfant ne mange pas trop d’aliment à haute teneur en gras. RST1c: Je dois m’assurer que mon enfant ne mange pas trop de ses aliments préférés. RST2: Je garde intentionnellement certains aliments hors de portée de mon enfant. RST3a: J’offre des sucreries (bonbons, crème glacée, gâteau, pâtisserie) à mon enfant pour le récompenser d’un bon comportement. RST3b: J’offre à mon enfant ses aliments préférés en échange d’un bon comportement. RST4a: Si je ne guidais pas ou que je ne contrôlais pas l’alimentation de mon enfant, il/elle mangerait trop de malbouffe (pizza, frites, poutines, hot dog). RST4b: Si je ne guidais pas ou que je ne contrôlais pas l’alimentation de mon enfant, il/elle mangerait trop de ses aliments préférés.	1 = Disagree 2 = Slightly disagree 3 = Neutral 4 = Slightly disagree 5 = Agree	1 = En désaccord 2 = Un peu en désaccord 3 = Neutre 4 = Un peu en accord 5 = En accord
6. Pressure to eat (PE)	6. Pression pour manger		
PE1: My child should always eat all the food in his/her plate. PE2: I have to be especially careful to make sure my child eats enough. PE3: If my child says “I am not hungry”, I try to get him/her to eat anyway. PE4: If I did not guide or regulate my child’s eating, he/she would eat much less than he/she should.	PE1: Mon enfant devrait toujours manger tout ce qu’il y a dans son assiette. PE2: Je dois porter une attention particulière pour m’assurer que mon enfant mange assez. PE3: Si mon enfant dit “Je n’ai pas faim”, j’essaie de le/la faire manger quand même. PE4: Si je ne guidais pas ou que je n’encadrais pas l’alimentation de mon enfant, il/elle mangerait beaucoup moins qu’il/elle le devrait.	1 = Disagree 2 = Slightly disagree 3 = Neutral 4 = Slightly disagree 5 = Agree	1 = En désaccord 2 = Un peu en désaccord 3 = Neutre 4 = Un peu en accord 5 = En accord
7. Monitoring (MN)	7. Surveillance		
MN1: How much do you keep track of the sweets (candy, ice cream, cake, pastries) your child eats? MN2: How much do you keep track of the snack food (potato chips, Doritos, or cheese puffs) your child eats? MN3: How much do you keep track of the high-fat foods your child eats?	MN1: À quelle fréquence surveillez-vous les sucreries (bonbons, crème glacée, gâteau, pâtisserie) que votre mange ? MN2: À quelle fréquence surveillez-vous les grignotines (chips, Doritos, crottes de fromage) que votre enfant mange ? MN3: À quelle fréquence surveillez-vous les aliments à haute teneur en gras que votre enfant mange ?	1 = Never 2 = Rarely 3 = Sometimes 4 = Mostly 5 = Always	1 = Jamais 2 = Rarement 3 = Parfois 4 = Souvent 5 = Toujours

CCW: Concern about Child’s Weight; MN: Monitoring; PCW: Perceived Child Weight; PE: Pressure to Eat; PPW: Perceived Parent Weight; PR: Perceived Responsibility; RST: Restriction.

**Table 3 behavsci-13-00487-t003:** Fit indices.

M	Description	χ^2^	df	AIC ^1^	BIC ^1^	CFI^2^	TLI ^2^	RMSEA ^3^	SRMR ^4^
1	7-factor, 25-item model	456.5 ***	257	10,278	10,549	0.731	0.686	0.077	0.098
2	7-factor, 21-item (3 composite items) model	232.4 **	171	8263	8242	0.889	0.864	0.052	0.070
3	7-factor, 21-item (3 composite items) model 1 error covariance	198.7 ^ns^	170	8229	8209	0.948	0.936	0.035	0.065
4	7-factor, 23-item (RST3a and RST3b removed) model	377.1 ***	212	9197	9175	0.764	0.718	0.078	0.096
5	7-factor, 20-item (2 composite items; RST3a and RST3b removed) model	204.0 **	152	7782	7762	0.903	0.878	0.051	0.069
**6**	**7-factor, 20-item (2 composite items; RST3a and RST3b removed) model** **1 error covariance**	**170.7 ^ns^**	**151**	**7748**	**7729**	**0.963**	**0.954**	**0.031**	**0.063**
7	8-factor, 25-item model	432.5 ***	251	10,266	10,242	0.754	0.706	0.075	0.094
8	8-factor, 22-item (2 composite items) model	247.0 **	185	8848	8825	0.891	0.864	0.050	0.070
9	8-factor, 22-item (2 composite items) model 1 error covariance	212.6 ^ns^	184	8814	8791	0.950	0.937	0.034	0.066

Bold characters indicate the best fitting model. ^1^ Lower values indicate better fit. ^2^ Values between 0.90 and 0.94 are indicative of an acceptable fit, and values ≥ are indicative of a good fit. ^3^ Values between 0.06 and 0.08 are indicative of an acceptable fit, and values < 0.06 are indicative of a good fit. ^4^ Values nearing 0.08 are indicative of a good fit. ^ns^
*p* ≥ 0.05; ** *p* < 0.01; *** *p* < 0.001. AIC: Akaike Information Criterion; BIC: Bayesian Information Criterion; CFI: Comparative Fit Index; df: Degrees of Freedom; RMSEA: Root Mean Square Error of Approximation; SRMR: Standardized Root Mean Square Residual; TLI: Tucker–Lewis Fit Index.

**Table 4 behavsci-13-00487-t004:** Mothers’ and children’s characteristics.

Mothers’ Characteristics (*n* = 136)	
Age (years) *	39.1 ± 5.7
Immigration status (%, yes)	94.1
Time since immigration (years) *^Ɨ^	8.8 ± 8.1
Region of birth (%)	
Sub-Saharan Africa	61.0
The Caribbean	33.1
Canada	5.9
Education level (%)	
Less than high school diploma	2.9
High school diploma or equivalent	8.1
College certificate or diploma	41.9
Undergraduate university certificate or bachelor’s degree	33.8
Graduate university certificate or degree	13.2
Matrimonial status (%)	
With partner	66.2
Single	33.8
BMI (kg/m^2^) *^ǂ^	30.0 ± 5.3
Weight status (%) ^ǂ^	
Underweight ^1^ and normal weight	17.7
Overweight	35.4
Obesity	46.9
**Children’s Characteristics (*n* = 136)**	
Age (years) *	8.8 ± 2.0
Sex (%, girls)	48.5
BMI z-score *^¥^	0.84 ± 1.1
Weight status (%) ^¥^	
Underweight and normal weight	64.9
Overweight	17.9
Obesity	17.2

* Value presented as Mean ± Standard deviation. ^1^ Only 1 mother had a BMI < 18.5 kg/m^2^. ^Ɨ^ *n* = 128; ^ǂ^ *n* = 130; ^¥^ *n* = 134.

**Table 5 behavsci-13-00487-t005:** Item and reliability measures.

Item	Mean	Median (IQR)	Kurtosis	Skewness	Cronbach α	Macdonald’s ω
PR1	4.58	5.00 (4.00–5.00)	2.25	−1.78	0.57	0.60
PR2	3.68	4.00 (3.00–5.00)	−0.89	−0.71		
PR3	4.50	5.00 (4.00–5.00)	4.84	−2.22		
PPW4	3.54	3.00 (3.00–4.00)	−0.35	0.39	-	-
PCW1	3.02	3.00 (3.00–3.00)	−0.55	3.62	0.68	0.78
PCW2	3.03	3.00 (3.00–3.00)	−0.41	5.54		
PCW3	2.94	3.00 (3.00–3.00)	−1.41	7.23		
CCW1	1.96	1.00 (1.00–3.00)	1.18	0.07	0.79	0.79
CCW2	2.15	1.00 (1.00–3.00)	0.95	−0.64		
CCW3	2.45	2.00 (1.00–4.00)	0.57	−1.32		
RST1A	4.50	5.00 (5.00–5.00)	−2.35	4.00	0.66	0.79
RST1B	4.15	5.00 (5.00–5.00)	−2.29	4.07		
RST1C	4.02	5.00 (3.75–5.00)	−1.23	0.19		
RST1.AV	4.34	4.67 (4.00–5.00)	−1.70	2.49		
RST2	3.65	5.00 (1.00–5.00)	−0.68	−1.45		
RST3A	2.16	1.00 (1.00–4.00)	0.82	−1.17		
RST3B	2.84	3.00 (1.00–5.00)	0.07	−1.85		
RST3.AV	2.50	2.50 (1.00–3.00)	0.40	−1.14		
RST4A	4.10	5.00 (4.00–5.00)	−1.32	−0.14		
RST4B	4.24	5.00 (4.00–5.00)	−1.57	0.77		
RST4.AV	4.17	5.00 (3.88–5.00)	−1.45	0.66		
PE1	3.63	4.00 (2.00–5.00)	−0.72	−1.24	0.64	0.70
PE2	4.46	5.00 (5.00–5.00)	−2.18	3.05		
PE3	3.34	4.00 (1.00–5.00)	−0.40	−1.66		
PE4	3.63	5.00 (1.00–5.00)	−0.67	−1.43		
MN1	3.99	4.00 (3.00–5.00)	−1.00	−0.06	0.80	0.82
MN2	3.88	4.00 (3.00–5.00)	−0.93	−0.13		
MN3	4.03	4.00 (3.00–5.00)	−1.14	0.26		

CCW: Concern about Child’s Weight; IQR: Interquartile Range; MN: Monitoring; PCW: Perceived Child Weight; PE: Pressure to Eat; PPW: Perceived Parent Weight; PR: Perceived Responsibility; RST: Restriction.

**Table 6 behavsci-13-00487-t006:** Standardized factor loadings and subscales’ factor–factor correlations.

CFQ Subscales and Items	PR	PPW	PCW	CCW	RST	PE	MN
PR	1.000	0.006	−0.187	0.015	0.236	0.283 *	0.374 *
PR1	0.58 **						
PR2	0.48 **						
PR3	0.64 **						
PPW		1.000	−0.147	0.028	0.079	−0.260 *	−0.001
PCW			1.000	0.179	−0.023	−0.072	0.015
PCW1			0.40 **				
PCW2			0.62 **				
PCW3			0.70 **				
CCW				1.000	0.237	0.044	0.283 **
CCW1				0.73 ***			
CCW2				0.72 ***			
CCW3				0.80 ***			
RST					1.000	0.361 **	0.613 **
RST1					0.47 **		
RST2					0.51 **		
RST4					0.57 **		
PE						1.000	0.330 **
PE1						0.30 **	
PE2						0.53 **	
PE3						0.69 **	
PE4						0.79 **	
MN							1.000
MN1							0.85 ***
MN2							0.79 ***
MN3							0.66 ***

*** <0.001. ** <0.01. * <0.05. CCW: Concern about Child’s Weight; CFQ: Child Feeding Questionnaire; MN: Monitoring; PCW: Perceived Child Weight; PE: Pressure to Eat; PPW: Perceived Parent Weight; PR: Perceived Responsibility; RST: Restriction.

## Data Availability

The data presented in this study are available on request from the corresponding author. The data are not publicly available due to privacy restrictions.

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
