# Peer review of "Assessing the Validity and Reliability of a French Version of the Child Feeding Questionnaire among Black Mothers Living in Ottawa (Canada)"

_behavsci, 2023, doi:10.3390/bs13060487_

Round 1
Reviewer 1 Report
The manuscript entitled “Assessing the validity and reliability of a French version of the Child Feeding Questionnaire among Black mothers living in Ottawa (Canada)” has aim to assess the construct validity and reliability of the French version of the CFQ among Black mothers of school-aged children living in Canada (Ottawa)
The topic is interesting because we know that nutrition and dietary habits of children are crucial for health and prevention of cardiovascular diseases later in life, also it is important because of the expansion of research and the application of tools in another language.
Some of the comments and suggestions for the manuscript:
Abstract:
Line 11 – „…Canadian studies assessing its construct validity“– why is so important that there is Canadian version, how it differs from USA population (especially when it is about black population)
Introduction:
Line 53 - For over twenty years, research on child feeding practices has primarily relied on the Child Feeding Questionnaire - very bold sentence because researches are carried out with the help of many different validate tools (including questionaries).
Materials and Methods – Participants
Quite a small number of respondents considering the number of inhabitants and children in Qttawa. Can you provide information on the total number of children aged 6-12 in Ottawa who meet the inclusion criteria? What percentage of possible respondents did you cover?
Please, explain difference between n=186 in paper ref 40, compared to n=136 in this study.
You have some overlaps with data from mentioned reference
Concept and presentation of results is very similar to ones from Birch et. al. (2021), therefore, it should be made more clear what is new and have scientific soundness as well as how to distinguish that presented paper is more than a professional work (not only validation of the translation itself).
Literature:
Why you didn’t include other relevant paper such as Papaioannou, M.A.; Power, T.G.; O’Connor, T.M.; Fisher, J.O.; Micheli, N.E.; Hughes, S.O. Child Weight Status: The Role of Feeding Styles and Highly Motivated Eating in Children. Children 2023, 10, 507.
Reviewer 2 Report
Thank you for the opportunity to review this interesting work. The authors have aimed to assess the reliability and construct validity of a French version of the CFQ among Black mothers of school-aged children living in Ottawa, Canada. The findings showed that the 7-factor model with minor modifications was the best fit for the current data.
Overall, the authors have a well-designed paper to validate the CFQ. They selected Black mothers and children as they are at a higher risk for overweight and obesity. An overall comment, it would be helpful if the authors provide a clearer rationale regarding why they decided to validate the French version of the CFQ.
Minor Comments:
· The authors should mention the rationale for selecting Black mothers and children in the paragraph of the introduction where they list the study objective. Although the beginning of the introduction has these details it will help to present a clearer rationale for selecting this group.
· This is a cross-sectional study but have other studies been done where the CFQ was administered over time as eating and feeding behaviors change across time.
· Why did the study exclude mothers who had adopted children? Assuming it is due to genetics being a risk factor for overweight and obesity. It will help if this is clarified.
· Lines 98-99: did 252 mothers complete the questionnaire or were 252 recruited and only 141 completed the questionnaire? Please clarify.
· It would be interesting to see similar studies using populations from different regions of Canada and across diverse racial and ethnic groups.
· For the tables and figures it will be helpful to include a reference to table 2 to understand all the acronyms. There are too many acronyms which makes it hard to follow.
Author Response
We have uploaded a document with our answers to the comments of the reviewer. Thank you for consulting the attached document.

Round 2
Reviewer 1 Report
I would like to thank the authors for their efforts to improve the manuscript.
Some parts of the manuscript are now clearer. Some new references have been added and rearrangements have been made to the text.